# The Resistance Levels and Target-Site Based Resistance Mechanisms to Glyphosate in *Eleusine indica* from China

**Jinyao Li, Yu Mei, Lingling Zhang, Lubo Hao and Mingqi Zheng ***

Department of Applied Chemistry, College of Science, China Agricultural University, Beijing 100193, China
* Correspondence: mqzheng@cau.edu.cn

**Abstract:** The Dongting and Poyang Lakes are the important rice growing areas, and the Bohai Rim and Loess Plateau are the main producing areas of apples in China, where glyphosate has been used continuously to control weeds including *Eleusine. indica* for many years. In this study, the resistance levels and target-site based resistance (TSR) mechanisms to glyphosate in *E. indica* populations, which were collected from above areas were investigated. A total of 35 out of 50 (70%) *E. indica* populations have evolved resistance to glyphosate with resistance index (RI) of 2.01~10.43. The glyphosate-resistant (GR) *E. indica* accumulated less shikimic acid than glyphosate-susceptible (GS) populations, when treated by 1.0 mg/L, 10 mg/L or 100 mg/L glyphosate. There was no mutation at Thr102 and Pro106 in 5-enolpyruvate shikimate-3-phosphate synthase (EPSPS), which endowed glyphosate resistance in *E. indica* and other weed species. A Pro-381-Leu was found in EPSPS in GR populations. In contrast, the expression level of *EPSPS* gene was highly correlated with glyphosate resistance in *E. indica* with a determination coefficient of 0.88. These indicate that the glyphosate resistance in aforementioned *E. indica* populations was mainly caused by the overexpression of EPSPS, not by amino acid mutation in EPSPS.

**Keywords:** glyphosate; *Eleusine indica*; shikimic acid; 5-enolpyruvate shikimate-3-phosphate synthase (EPSPS)





## 1. Introduction

Glyphosate is considered a once-in-a-century herbicide, and is used for nonselective controlling of the majority of weeds in agricultural and nonagricultural situations. It kills weeds by inhibiting 5-enolpyruvate shikimate-3-phosphate synthase (EPSPS), which is a key enzyme in the biosynthesis of the aromatic amino acids tryptophan (Try), tyrosine (Tyr) and phenylalanine (Phe) [1]. The glyphosate inhibits EPSPS by competing with phosphoenolpyruvate for the binding site. In addition, the EPSPS only occurs in plants and microorganisms, so glyphosate displays low toxicity to animals [2].

Overreliance on glyphosate has made more and more weeds evolve resistance to glyphosate since the middle of 1990s [3]. The glyphosate resistance was first reported in *Lolium rigidum* L. from Australia in 1998, which was about 20 years after glyphosate commercialization [4]. To date, at least 56 weed species in 28 different countries have evolved resistance to glyphosate [5]. The glyphosate resistance can be endowed by the target-site based (TSR) mechanisms due to amino acid mutation or overexpression of the EPSPS. The resistance mutation has been documented at the site of Pro106 (changing Pro106 to any of Ala/Leu/Ser/Thr) and Thr102 in EPSPS, which usually causes weeds in field to evolve moderate resistance to glyphosate [6,7]. A double amino acid mutation (Thr-102-Ile + Pro-106-Ser) was identified in glyphosate-resistant (GR) *E. indica*, which showed high level resistance to glyphosate [8]. Recently, a triple amino acid mutation (Thr-102-Ile + Ala-103-Val+Pro-106-Ser) was found in GR *Amaranthus hybridus* [9]. In addition, the over-expression of the *EPSPS* gene was reported in GR *L. rigidum* and *Conyza* spp. [10,11], and *EPSPS* gene amplification was demonstrated in GR *Amaranthus palmeri*, *Amaranthus tuberculatus*, *Kochia*

*scoparia*, *Lolium perenne* ssp. *multiflorum*, *Chloris truncata* and *Bromus diandrus* et al. [12,13]. In contrast to the TSR mechanisms, the glyphosate resistance can be conferred by multiple non-target-site based (NTSR) mechanisms. For example, reduced glyphosate foliar uptake and/or translocation in GR *Sorghum halepense* and *L. rigidum*, vacuolar sequestration and enhanced metabolism by an aldo-keto reductase in GR *Echinochloa colona* [14–19].

Goosegrass (*E. indica*) is an annual, self-pollinating and diploid grass specie. It is considered one of the "world's worst weeds" and has been reported to be a problematic weed in a large number of crops all over the world [10]. *E. indica* has been reported to evolve resistance to the herbicides glyphosate, glufosinate-ammonium, trifuluralin, paraquat, clethodim, cyhalofop-butyl, fluazifop-butyl, haloxyfop-methyl, sethoxydim, imazapyr et al. [5]. Just as in other countries, *E. indica* is also a notorious weed and widely distributed in both crop and non-crops fields in China. The *E. indica* also have evolved resistance to various herbicides. For example, the *E. indica* in direct-seeded rice showed resistance to metamifop, cyhalofop-butyl, sethoxydim, haloxyfop-R-methyl and fenoxaprop-p-ethyl [20]. The *E. indica* from cotton fields in the Hunan province was reported to develop resistance to quizalofop-p-ethyl [21]. Although the glyphosate has been used to control weeds in orchards, plantations and other non-crop fields for more than forty years, the extensive investigations on glyphosate resistance in *E. indica* have not been carried out. In this study, the glyphosate resistance level of *E. indica* infesting farmland around Dongting (DT) and Poyang (PY) lakes, apple fields around Bohai Rim and Loess Plateau was investigated. The shikimic acid accumulation and relative expression level of *EPSPS* mRNA were determined. In addition, the amino acid mutation of EPSPS in GR and glyphosate-susceptible (GS) populations was identified.

## 2. Materials and Methods

### 2.1. Plant Materials

The *E. indica* seeds were collected from apple orchards in provinces of Beijing (BJ), Hebei (HB), Liaoning (LN) and Shandong (SD) around Bohai Rim; provinces of Henan (HN), Shanxi (SX) and Shannxi (SHX) around Loess Plateau; and from farmlands around Dongting (DT) and Poyang (PY) lakes in China, where glyphosate was used to control *E. indica* for many years (Figure 1, Table S1).

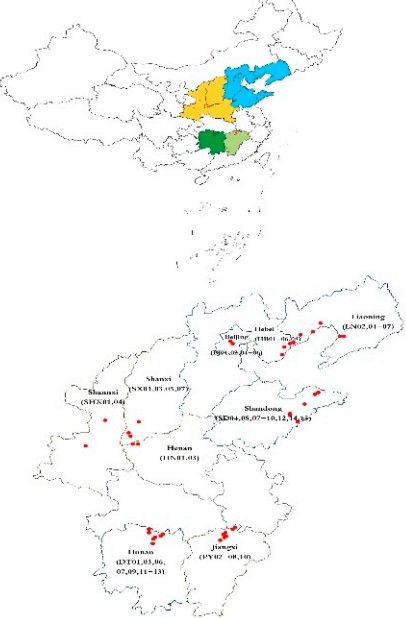

**Figure 1.** The geographical origins of different *E. indica* populations from apple orchards around Bohai Rim (Beijing, BJ; Hebei, HB; Liaoning, LN; Shandong, SD) and Loess Plateau (Henan, HN; Shanxi, SX; Shannxi, SHX) farmland around Dongting (DT) and Poyang (PY) lakes in China.

Seeds were germinated on a filter paper in a Petri dish, then transplanted into pots containing moist loam soil and grown in an artificial climate chamber. The conditions of germination and growth in the chamber were 25 °C/20 °C (light/dark), 16 h photoperiod and luminous intensity of about 15,000~20,000 Lux.

### 2.2. Petri Dish Dose-Response Bioassay

The test concentration of glyphosate was 0.1, 0.5, 2.5, 12.5, 62.5, 125 or 250 mg/L, respectively. Firstly, 5.0 mL glyphosate solution was added into each Petri dish containing one layer of filter paper. Then, germinated *E. indica* seeds were placed evenly in each Petri dish, and grown in an artificial climate chamber with the same conditions described in Section 2.1. The root length of each plant was measured 6 days later, and the $GR_{50}$ value (herbicide rate causing 50% growth reduction of the root lengths) was estimated by logistic regression. Three replications were set for each glyphosate concentration. The experiment was repeated at different times.

### 2.3. Extraction and Accumulation Assay of Shikimic Acid

The shikimic acid was measured using the leaf disc method described by Mei et al. (2018) [11]. First, the Petri dish was filled with 5 mL $NaH_2PO_4$ solution (pH 4.4) containing 1.0, 10 or 100 mg/L glyphosate. Next, two leaves with a length of 5 mm were cut from each of the five *E. indica* plants and immersed into glyphosate solutions. The Petri dishes were transferred into an artificial climate chamber for 16 h under the conditions of 25 °C, continuous 16 h illumination. All leaves were ground in liquid nitrogen, and then 1 mL of 1.25 M HCl was added to extract the shikimate. Finally, the samples were incubated at 37 °C with agitation for 30 min after mixing vigorously for 5 min, and then centrifuged at 12,000 g for 10 min. The supernatants were collected and used for shikimic acid determination.

The reaction mixtures were incubated at 37 °C for 60 min, which contained 250 μL supernatants, 1.0 mL 0.25% ($w/v$) periodic acid and 0.25% ($w/v$) sodium metaperiodate. The reaction was stopped by adding 1 mL 0.6 M NaOH and 0.22 M $Na_2SO_3$. The optical density (OD) at 383 nm was determined colorimetrically using microplate photometer (BioTek, Winooski, VT, USA). Six replicates were set for each glyphosate concentration. The shikimic acid concentration was calculated based on a standard curve. The experiments were repeated four times with independent extracts. The shikimic acid concentration was expressed in μg $cm^{-2}$ leaves.

### 2.4. RNA Extraction and cDNA Synthesis

Pooled samples of six plants at the stage of 6-leaf were used for RNA extraction. Total RNA was extracted according to the instruction of RNApre Pure Plant Kits (Tiangen, Beijing, China). First-strand cDNA was reverse-transcribed using 2 μg total RNA (Tiangen, Beijing, China). The cDNA was synthesized in a 25 μL reaction mixture, which included 1 μL cDNA, 0.5 μL primer pairs, 12.5 μL 2 × PCR Mix and 10.5 μL ddH$_2$O. The primer pairs (5′-GGTGGATAACCTTTTAAACAGTGAG-3′ and 5′-TTAGTTCTTGACGAAAGTGCTGAG-3′) were designed according the EPSPS sequence of *E. indica* (HQ403647). The PCR programs consisted of one denaturation step of 3 min at 94 °C, 30 cycles of 30 s at 94 °C, 30 s at 56 °C, 72 °C for 30 s, followed by a final extension step of 5 min at 72 °C.

### 2.5. Identification of Mutation in EPSPS

The purified PCR product was cloned into the pLB vector (Tiangen, Beijing, China), and then transformed into TOP10 Chemically Competent Cells (Tiangen, Beijing, China). The plasmids containing the insert were extracted and sequenced directly with the primer pairs (5′-CGACTCACTATAGGGAGAGCGGC-3′ and 5′-AAGAACATCGATTTTCCATGGCAG-3′). Six clones were selected for sequencing. Ant potential resistance mutation was identified by comparing the EPSPS sequence with that of wild *E. indica* specie.

### 2.6. EPSPS Gene Expression Level

The expression level of the *EPSPS* gene in *E. indica* was determined using Super Real Pre Mix Plus (SYBRGreen). The primer pairs of EPSPS (5′-CTGATGGCTGCTCCTTTAGCTC-3′ and 5′-CCCAGCTATCAGAATGCTCTGC-3′) and reference gene *β-actin* (5′-AACAGGG-AGAAGATGACCCAGA-3′ and 5′-GCCCACTAGCGTAAAGGGACAG-3′) were designed. The reaction mixtures contained 10 μL 2 × Super Real Pre Mix Plus, 0.6 μL primers (0.1 μM), 1 μL diluted cDNA (25 ng/μL), 0.4 μL 50 × ROX Reference Dye and 7.4 μL RNase-free ddH$_2$O. The RT-qPCR program started with an incubation at 95 °C for 15 min, followed by 40 cycles of 95 °C for 10 s, 60 °C for 20 s and 72 °C for 32 s. Under this condition, the internal reference and the target gene displayed high amplification efficiency (92~100%). Three biological replicates and four technical replicates were performed for each *E. indica* population.

The relative expression ratio (as $2^{-\triangle\triangle C_T}$) was calculated by the comparative CT method, where $\triangle C_T$ = [$C_T$ target gene − $C_T$ internal control gene] and $\triangle\triangle C_T$ = [$\triangle C_T$ resistant − $\triangle C_T$ susceptible] [22].

### 2.7. Statistical Analysis

Dose-response curves and GR$_{50}$ (herbicide rate causing 50% growth reduction of plants) were obtained by a non-linear regression using the log-logistic Equation (1) [23]:

$$y = C + (D - C)/(1 + (x/GR_{50}))^b \qquad (1)$$

where y represents shoot biomass (percentage of control) at herbicide rate *x*; C and D are the lower and upper limits, respectively; and b is the slope at GR$_{50}$. The expression difference of the *EPSPS* gene between GR and GS (BJ05) *E. indica* populations was analyzed by an independent-sample T test at *p* = 0.05 or 0.01 significance level.

## 3. Results

### 3.1. Glyphosate Resistance of E. Indica

The sensitivity of 50 *E. indica* populations (34 populations from Bohai Rim and Loess Plateau, and 16 populations Dongting and Poyang lakes) to glyphosate was determined. The results indicated that 35 out of 50 (70%) populations have developed resistance to glyphosate according to the resistance standard of an RI value greater than 2.0. The proportion of *E. indica* populations with an RI value of 2~4 and 4~10 is 32% and 36%, respectively. One (2%) population (SX03) evolved a high resistance level to glyphosate with an RI value of 10.42. Among them, 81.25% of the populations around Dongting and Poyang Lakes have evolved resistance to glyphosate, while the proportion of GR populations around Dongting (DT) and Poyang (PY) Lakes is 64.7% (Figure 2, Table S2).

### 3.2. Shikimate Accumulation

In this study, the shikimate accumulation in 27 *E. indica* populations was tested. The results indicated that the GR populations accumulated less shikimate than GS populations, when *E. indica* was treated by glyphosate at concentrations of 1.0 mg/L, 10 mg/L or 100 mg/L (Table 1). The Pearson coefficient between the GR$_{50}$ and shikimate concentration was −0.73, −0.74 and −0.69, when the *E. indica* plants were treated by glyphosate with concentrations of 1.0 mg/L, 10 mg/L or 100 mg/L, respectively. This indicated that the level of shikimate had a significant negative relationship with *E. indica* resistance to glyphosate. In addition, all *E. indica* populations can accumulate more shikimate with the increasing of glyphosate concentration (Table 1).

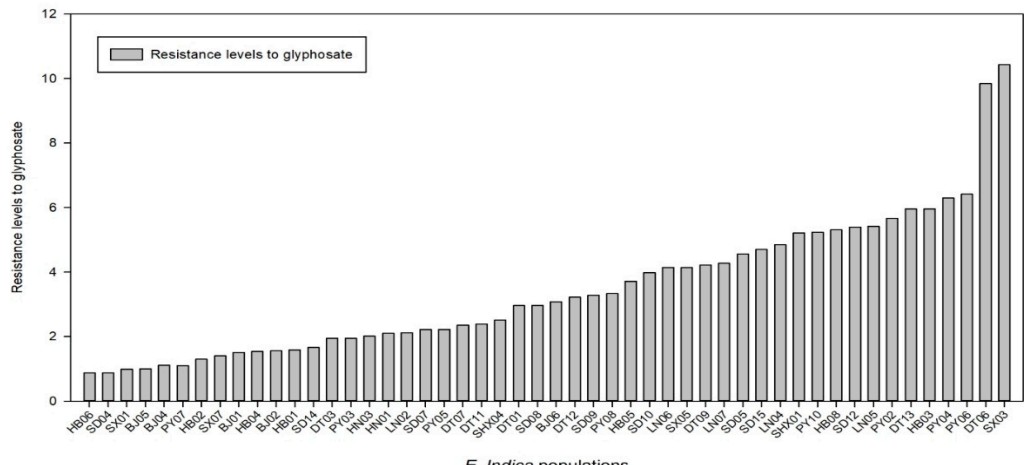

**Figure 2.** Glyphosate resistance level of different *E. indica* accessions from farmland around Dongting (DT) and Poyang (PY) lakes, apple orchards around areas of Bohai Rim (Beijing, BJ; Hebei, HB; Liaoning, LN; Shandong, SD) and Loess Plateau (Henan, HN; Shanxi, SX; Shannxi, SHX) in China.

**Table 1.** Shikimate accumulation in *E. indica* leaves treated by glyphosate at concentrations of 1.0, 10 and 100 mg/L. Data were sorted from largest to smallest according the shikimic acid concentration treated by 100 mg/L glyphosate.

| Populations | Glyphosate (mg/L) | | | | | | |
| --- | --- | --- | --- | --- | --- | --- | --- |
| | 0 (CK) | 1.0 | | 10 | | 100 | |
| | Shikimic Acid | | | | | | |
| | ($\mu g/cm^2$) | ($\mu g/cm^2$) | R/S | ($\mu g/cm^2$) | R/S | ($\mu g/cm^2$) | R/S |
| BJ05 | 18.30 ± 2.39 | 21.88 ± 1.15 | 1.00 | 23.55 ± 1.25 | 1.00 | 57.76 ± 7.73 | 1.00 |
| PY07 | 14.12 ± 1.93 | 24.92 ± 1.39 * | 1.14 | 46.96 ± 2.46 *** | 1.99 | 56.23 ± 2.10 | 0.97 |
| SD04 | 26.74 ± 1.23 | 29.02 ± 2.07 ** | 1.33 | 36.17 ± 3.57 ** | 1.54 | 53.35 ± 3.32 | 0.92 |
| BJ01 | 25.91 ± 3.66 | 31.00 ± 1.74 *** | 1.42 | 36.17 ± 4.27 ** | 1.54 | 52.66 ± 3.12 | 0.91 |
| HN01 | 11.08 ± 1.17 | 15.72 ± 1.67 ** | 0.72 | 23.09 ± 5.74 | 0.98 | 41.56 ± 2.80 * | 0.72 |
| HN03 | 10.78 ± 1.30 | 19.67 ± 0.82 * | 0.9 | 25.68 ± 2.57 | 1.09 | 40.65 ± 1.24 * | 0.7 |
| LN02 | 22.03 ± 2.17 | 24.01 ± 3.27 | 1.1 | 26.51 ± 1.12 * | 1.13 | 40.65 ± 6.08 * | 0.7 |
| SHX04 | 27.12 ± 2.60 | 31.07 ± 4.30 * | 1.42 | 33.20 ± 0.53 *** | 1.41 | 39.51 ± 3.11 * | 0.68 |
| DT01 | 28.72 ± 3.52 | 29.02 ± 3.09 | 1.33 | 34.04 ± 1.17 *** | 1.45 | 37.23 ± 4.24 ** | 0.64 |
| SD08 | 23.02 ± 1.63 | 25.22 ± 0.63 * | 1.15 | 28.19 ± 1.24 ** | 1.2 | 36.40 ± 4.43 ** | 0.63 |
| DT11 | 9.26 ± 0.67 | 12.53 ± 2.14 *** | 0.57 | 13.06 ± 0.87 *** | 0.55 | 34.49 ± 1.65 ** | 0.6 |
| DT09 | 14.73 ± 1.30 | 20.13 ± 2.01 | 0.92 | 26.36 ± 2.45 | 1.12 | 33.51 ± 2.88 ** | 0.58 |
| PY08 | 19.22 ± 0.18 | 21.42 ± 2.25 | 0.98 | 31.30 ± 0.87 *** | 1.32 | 31.91 ± 3.66 ** | 0.55 |
| SD10 | 9.64 ± 0.66 | 13.90 ± 1.38 *** | 0.64 | 15.11 ± 0.96 *** | 0.64 | 30.54 ± 1.91 ** | 0.53 |
| LN07 | 18.99 ± 1.71 | 19.22 ± 1.82 * | 0.88 | 20.05 ± 1.91 * | 0.85 | 27.27 ± 3.62 ** | 0.47 |
| SD05 | 17.24 ± 2.37 | 17.62 ± 3.76 | 0.81 | 19.06 ± 3.47 | 0.81 | 26.29 ± 2.28 ** | 0.46 |
| HB08 | 11.84 ± 0.91 | 13.59 ± 2.03 *** | 0.62 | 20.74 ± 3.69 | 0.88 | 24.16 ± 4.18 ** | 0.42 |
| SX05 | 11.61 ± 0.95 | 13.06 ± 1.90 *** | 0.6 | 14.35 ± 1.35 *** | 0.61 | 22.71 ± 2.64 ** | 0.39 |
| PY06 | 8.50 ± 2.05 | 10.17 ± 1.32 *** | 0.46 | 11.46 ± 0.25 *** | 0.49 | 21.88 ± 6.39 *** | 0.38 |
| LN04 | 10.02 ± 0.67 | 12.45 ± 1.00 *** | 0.57 | 15.34 ± 0.91 *** | 0.65 | 21.57 ± 1.90 ** | 0.37 |
| LN05 | 12.98 ± 0.56 | 12.98 ± 1.08 *** | 0.59 | 14.73 ± 3.12 ** | 0.63 | 19.82 ± 6.56 *** | 0.34 |
| SX03 | 8.95 ± 1.30 | 9.11 ± 0.38 *** | 0.42 | 9.26 ± 0.67 *** | 0.39 | 19.29 ± 4.24 *** | 0.33 |
| DT13 | 17.32 ± 0.52 | 17.01 ± 1.00 ** | 0.78 | 18.23 ± 1.23 ** | 0.77 | 18.76 ± 1.51 ** | 0.32 |
| PY02 | 11.08 ± 1.67 | 12.38 ± 1.14 *** | 0.57 | 16.18 ± 1.83 ** | 0.69 | 18.08 ± 1.03 ** | 0.31 |
| HB03 | 7.81 ± 0.25 | 9.26 ± 2.28 *** | 0.42 | 12.68 ± 1.79 *** | 0.54 | 17.01 ± 1.99 ** | 0.29 |
| SD12 | 9.94 ± 0.43 | 11.46 ± 0.96 *** | 0.52 | 11.84 ± 0.91 *** | 0.5 | 15.34 ± 1.83 ** | 0.27 |
| SHX01 | 8.35 ± 0.29 | 10.55 ± 0.25 *** | 0.48 | 10.93 ± 1.54 *** | 0.46 | 11.92 ± 0.72 ** | 0.21 |

Data are means ± SE of four replicates of four plants. The *, ** or *** indicated, respectively, the means of shikimic acid are significantly different at *p* = 0.05, 0.01 or 0.001 significance level compared to that of the susceptible (BJ05) biotype.

*3.3. EPSPS Gene Expression*

The results indicated that the expression level of the *EPSPS* gene in the GR *E. indica* population of DT11, PY08, SX05, SD05, DT09, LN07, SHX01, HB08, SD12, LN05, DT13, HB03, PY06 and SX03 was significantly higher than the GS population (BJ05). However, the EPSPS expression level of SD04, BJ01, PY07, HN03, LN02, HN01, SHX04, DT01 and SD08 showed no obvious difference with the BJ05 population. The *EPSPS* expression had a significant positive relationship with the resistance index (RI) of *E. indica* to glyphosate ($R^2 = 0.88$), which demonstrated the *EPSPS* overexpression conferring the *E. indica* resistance to glyphosate (Figure 3).

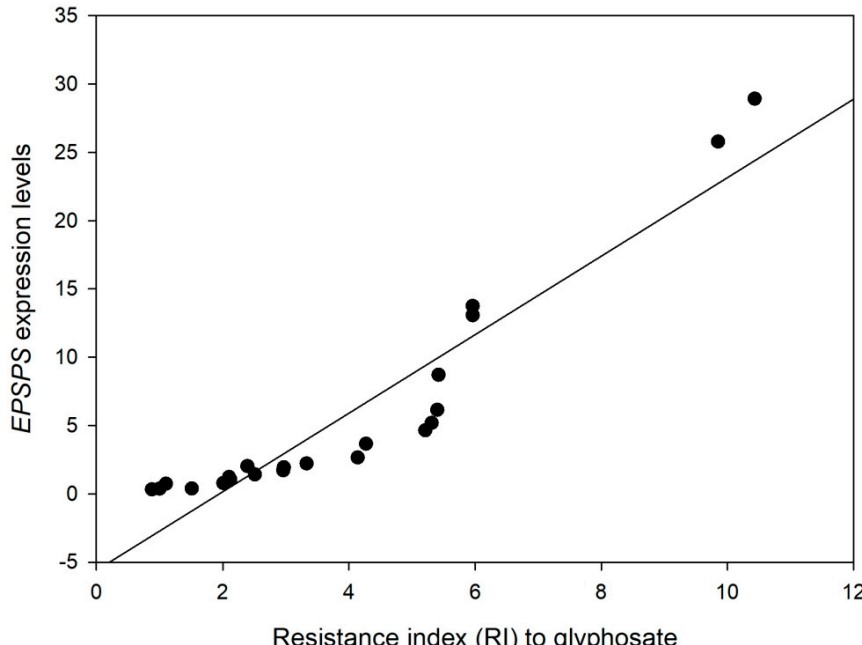

**Figure 3.** Linear regression analysis between the *EPSPS* expression level and resistance index (RI) to glyphosate in partial *E. indica* populations from farmland around Dongting and Poyang lakes, and apple orchards around areas of Bohai Rim and Loess Plateau in China.

*3.4. EPSPS Gene Sequencing*

The *E. indica* populations with different resistance levels to glyphosate from different areas were selected for *EPSPS* gene sequencing. The results indicated that there was no amino acid mutation at the site of Thr102 and Pro106 in EPSPS, which was reported to endow glyphosate resistance in *E. indica* and other weed species. It was interesting that a Pro381 Leu mutation was found in GR *E. indica* populations (HN01, PY08, DT09, SHX01, HB08, DT13, PY06 and SX03), but not in GS populations (SX01, PY07 and BJ04). However, it is not clear whether the Pro381 Leu mutation mediates GR *E. indica* resistance to glyphosate.

## 4. Discussion

The whole-plant experiments are usually used for the bioassay of herbicide efficacy to weeds. The results obtained by this method can better reflect the true effects of herbicides in the fields. However, this method is expensive, time-consuming and takes up a lot of space to test a large number of samples. The Petri dish is a simple, quick, reliable and inexpensive method that has been used before to identify GR *L. multiflorum* and *L. rigidum* [24–27]. Hence, the Petri dish bioassay was selected to test the glyphosate resistance of *E. indica* populations in this study due to huge *E. indica* populations.

At present, the weed of *E. indica*, *C. canadensis*, *Digitaria sanguinalis*, *Amaranthus retroflexus* and *Alternanthera philoxeroides* in China have evolved resistance to glyphosate due to the extensive and intensive use of glyphosate [11,28,29]. Dongting (DT) and Poyang (PY)

Lakes are the important planting areas of rice, and the areas around Bohai Rim and Loess Plateau are the main producing areas of apples in China, where glyphosate was widely used for many years. In this study, the glyphosate resistance levels of *E. indica* populations from the above areas were investigated. The results indicated that 81.25% of populations in farmland around Dongting and Poyang Lakes, and 64.7% of populations in apple orchards around Bohai Rim and Loess Plateau have evolved resistance to glyphosate. Among them, 37.5% and 43.75% populations around DT and PY lakes evolved low ($2 \leq RI < 4$) and moderate ($4 \leq RI \leq 10$) resistance levels to glyphosate, respectively. Additionally, 29.4%, 32.4% and 2.9% of populations around Bohai Rim and Loess Plateau evolved low, moderate and high ($RI > 10$) resistance levels to glyphosate, respectively (Figure 2). Overall, the proportion of GR populations around DT and PY lakes was higher than that around Bohai Rim and Loess Plateau, which may be due to the difference of *E. indica* samples from two areas. In addition, farmers living around Bohai Rim and Loess Plateau began to stop using glyphosate to control weeds in apple orchards in recent years, because the apple trees can suffer from glyphosate absorbed through exposed roots.

Glyphosate caused the shikimate accumulation in plants by inhibiting EPSPS activities. The GR weeds tend to accumulate less shikimate than GS populations, which was confirmed in various weed species, such as *C. truncata*, *Amaranthus spinosus*, *L. perenne* ssp. *Multiflorum*, *Hordeum glaucum* and *C. canadensis* [11,30–33]. Hence, the accumulation of shikimate is an important indicator of whether weed species are sensitive to glyphosate. Similar results were also obtained in GR *E. indica* populations from farmland around Dongting (DT) and Poyang (PY) lakes, apple orchards around areas of Bohai Rim and Loess Plateau. Obviously, the higher the resistance level to glyphosate, the less shikimic acid accumulated (Table 1). It indicated that the GR *E. indica* populations likely contained one or more TSR mechanisms, such as gene mutation, over expression or amplification. The results of *EPSPS* gene sequencing showed that there was no amino acid mutation at Thr102 or Pro106 in EPSPS, which was reported to confer glyphosate resistance in *E. indica* and other weed species [5,6,10]. In contrast, the *EPSPS* gene overexpressed in most GR *E. indica* populations, and the expression level of EPSPS was highly correlated with glyphosate resistance levels in *E. Indica* (Figure 3). Hence, we speculated that the over expression of EPSPS was mainly responsible for the glyphosate resistance of *E. indica* from areas around Bohai Rim, Loess Plateau, Dongting and Poyang lakes in China. It was interesting that a Pro381 Leu mutation was found in GR *E. indica* populations (HN01, PY08, DT09, SHX01, HB08, DT13, PY06 and SX03), but not in GS populations (SX01, PY07 and BJ04). The Pro381 Leu was also found in the *E. indica* population from Malaysia [10]. The X-ray crystallography of *E. coli* EPSPS, shikimate-3-phosphate and glyphosate ternary complex demonstrated that the residue at position 381 would likely reside on the EPSPS enzyme's outer surface within a turn between two beta-sheets, which was away from the active site of EPSPS. Thus, it is unlikely to be directly involved in catalysis, substrate binding, or domain closure [2,10]. In addition, *EPSPS* gene amplification, which is a common mechanism conferring glyphosate resistance, cannot be excluded.

## 5. Conclusions

The *E. indica* populations, which infested farmland around Dongting and Poyang Lakes, and apple orchards around Bohai Rim and Loess Plateau, have evolved resistance to glyphosate. Therefore, the glyphosate resistance in *E. indica* was mainly caused by over expression of the *EPSPS* gene, but not by gene mutation in *EPSPS*.

**Supplementary Materials:** The following supporting information can be downloaded at: https://www.mdpi.com/article/10.3390/agronomy12112780/s1, Table S1: The geographical origins of *E. indica* populations collected from apple orchards around Bohai Rim (Beijing, BJ; Hebei, HB; Liaoning, LN; Shandong, SD) and Loess Plateau (Henan, HN; Shanxi, SX; Shannxi, SHX), farmland around Dongting (DT) and Poyang (PY) lakes in China.; Table S2: The GR50 values estimated by dose-response curves of *E. indica* populations collected from apple orchards around Bohai Rim

(Beijing, BJ; Hebei, HB; Liaoning, LN; Shandong, SD) and Loess Plateau (Henan, HN; Shanxi, SX; Shannxi, SHX), farmland around Dongting (DT) and Poyang (PY) lakes in China.

**Author Contributions:** M.Z. conceived, designed and directed the research and revised the manuscript; J.L. collected seeds, performed most experiments and wrote the manuscript; Y.M. collected seeds, attended experiments and analyzed data; L.Z. and L.H. provided assistance during experiments and data analysis. All authors have read and agreed to the published version of the manuscript.

**Funding:** This research received no external funding.

**Institutional Review Board Statement:** Not applicable.

**Informed Consent Statement:** Not applicable.

**Data Availability Statement:** Not applicable.

**Conflicts of Interest:** The authors declare no conflict of interests.

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
