# Peer review of "The Resistance Levels and Target-Site Based Resistance Mechanisms to Glyphosate in Eleusine indica from China"

_agronomy, doi:10.3390/agronomy12112780_

Round 1

Reviewer 1 Report

The manuscript titled “The resistance levels and target-site based resistance mechanisms to glyphosate in Eleusine. indica from China” describes the characterization of several glyphosate-resistant populations of goosegrass in specific regions of China. The authors present the analysis of shikimate content in the different populations, at different concentration of glyphosate, The EPSPS gene was also sequence to locate potential resistance conferring mutations. Finally, the level of expression of the EPSPS gene was measured.

First, the manuscript would benefit if being reviewed with the help of someone competent in English e.g.: Total 34 E. indica populations (line 159), In present study (line 246), Obviously (line 248), Therefore, the results of EPSPS gene sequencing showed (lines 250-251), …

Second, results could be better presented. It would be nice to see a plot showing the RI index vs expression levels. Linking the same populations between the different experiments would be beneficial to the manuscript.

Finally, more information about the values obtained for doses-responses and expression level measurement should be provided. For example, authors have not mentioned the amplification efficiency of the different amplicons. Also, there is no information about the tissues collected, where they taken at the same stage, equivalent in quantity, what part of the plant was used?

Title: why a dot (“.”) after Eleusine in the title?

Line 13: 35 out of 70 is 50%

Lines 29-30: structure of sentence leads to believe that glyphosate-tolerant crops are non-agricultural. Moreover, are they tolerant or resistant?

Line 56: “world’s worst weeds”, what is the reference for this assertion?

Figure 1 is illegible, even if I zoom in.

Line 147: “[CT target gene –CT mean of two internal control genes]”. Only B-actin gene is mentioned as internal control.

Lines 161-162: 35 out of 70 is 50%

Figure 2, part of the figure is illegible, even if I zoom in. There is room for bigger font. Maybe you can save some space by putting all biotypes on the same graph.

Section 3.1, please provide more information about dose-response. It can be placed in supplemental results

Lines 181-182: which E. indica plants accumulated more shikimate?

Table 1: why is statistical test done only at 100 mg/L of glyhosate?

Paragraph starting at line 241: many species names are not italicized

Lines 253-255: “In contrast, the EPSPS gene overexpressed in most GR E. indica populations, and the expression level of EPSPS was highly correlated with glyphosate resistance in E. indica.” A graph of RI vs expression level would be very indicative of that assertion.

Lines 264-265: “In addition, EPSPS gene amplification, which is a common mechanism conferring glyphosate resistance, cannot be excluded.” It may actually explain the difference in expression.

Line 272: “Patents” There are no mention of patents. Did you mean “Acknowledgments”?

Species names in the references should be italicized.

Author Response

Response to reviewer 1

Reviewer 2 Report

The work is interesting and falls within the scope. However, there are several concerns, which need to be solved before acceptance. Please find my comments in the attached PDF.

Author Response

Response to reviewer 2

Round 2

Reviewer 1 Report

English can still be improved. 

Figure 3: why do you insist on a bar graph. Why not try a scatter plot with regression? Plus, you can also provide R2 for goodness of fit.

Expression level measurement: please provide reference for 2 square minus delta delta ct method.

Figure 1 quality should be increased even more.

Remove s at the end of "genes" since there is only one control gene
